# Extended Finite Element Method (XFEM) Model for the Damage Mechanisms Present in Joints Bonded Using Adhesives Doped with Inorganic Fillers

**DOI:** 10.3390/ma16237499

**Published:** 2023-12-04

**Authors:** João P. J. R. Santos, Daniel S. Correia, Eduardo A. S. Marques, Ricardo J. C. Carbas, Frida Gilbert, Lucas F. M. da Silva

**Affiliations:** 1Institute of Science and Innovation in Mechanical and Industrial Engineering (INEGI), University of Porto, Rua Dr. Roberto Frias 400, 4200-465 Porto, Portugaldcorreia@inegi.up.pt (D.S.C.); 2Department of Mechanical Engineering, Faculty of Engineering (FEUP), University of Porto, Rua Dr. Roberto Frias 400, 4200-465 Porto, Portugal; lucas@fe.up.pt; 3ArcelorMittal Global R&D, Rte de Saint-Leu, 60160 Montataire, France; frida.gilbert@arcelormittal.com

**Keywords:** XFEM, adhesive bonding, fracture mechanics, automotive industry, glass beads

## Abstract

The use of adhesive bonding in diverse industries such as the automotive and aerospace sectors has grown considerably. In structural construction, adhesive joints provide a unique combination of low structural weight, high strength and stiffness, combined with a relatively simple and easily automated manufacturing method, characteristics that are ideal for the development of modern and highly efficient vehicles. In these applications, ensuring that the failure mode of a bonded joint is cohesive rather than adhesive is important since this failure mode is more controlled and easier to model and to predict. This work presents a numerical technique that enables the precise prediction of the bonded joint’s behavior regarding not only its failure mode, but also the joint’s strength, when inorganic fillers are added to the adhesive. To that end, hollow glass particles were introduced into an epoxy adhesive in different amounts, and a numerical study was carried out to simulate their influence on single lap joint specimens. The numerical results were compared against experimental ones, not only in terms of joint strength, but also their failure pattern. The neat adhesive, which showed 9% and 20% variations in terms of failure load and displacement, respectively. However, looking at the doped configurations, these presented smaller variations of about 2% and 10% for each respective variable. In all cases, by adding glass beads, crack initiation tended to change from adhesive to cohesive but with lower strength and ductility, correctly modeling the general experimental behavior as intended.

## 1. Introduction

Of all the technological developments made in adhesive bonding technologies, structural adhesives became the most relevant for mechanical engineering, ensuring the structural integrity of the components to join, resisting much higher loads than the common non-structural adhesives and sealants [1,2,3]. This joining technique finds use in diverse applications, ranging from the electronics industry to the automotive and aerospace sectors. Bonded joints exhibit, generally, uniform stress distributions and provide excellent stiffness and strength to weight ratios [1,4]. In addition, this technique lends itself to industrial implementation in modern structures, since it is well suited for automated processes. In addition to being used to join parts with complex geometries composed of distinct designs, contours, thicknesses, perhaps most importantly, it is compatible with a wide range different materials [1,4].

Despite this growth, adhesive bonding is still a relatively new technique, and there is still a large interest in adopting methodologies and techniques that are suitable for enhancing the performance of bonded joints. The nature of these techniques is varied. Some are based on the application of functionally graded properties via temperature gradients, for example, the combination between adhesive joints and other conventional joining methods, known as hybrid joints; the geometrical optimization of the adherends’ configurations [1]; and the adoption of novel surface treatments, among many others [5].

Nevertheless, most of the approaches described provide limited improvements in return for greatly added process complexity. For those reasons, introducing reinforcements into the adhesive layer has become a popular and inexpensive alternative to improve adhesive joint performance [5].

Epoxy adhesives are widely used in structural applications, being amorphous and highly cross-linked materials with good strength and stiffness but suffering from a generally brittle behavior. Thus, the addition of a second phase of particles, fibers, or other kinds of reinforcements has a strong potential to enhance the adhesive’s mechanical behavior, depending on the reinforcement [6]. In terms of their nature, this can be ceramic [7,8], metallic [9,10,11], bio-based, carbon-based, silicone-based, among others [12,13,14]. Other than serving as reinforcements [8,9,10,15], i.e., improving the mechanical properties of the adhesive (stiffness, strength, toughness, fatigue, or others) fillers can have more uses. These applications can go from guaranteeing adhesive thickness, to thermal [11] or electrical [10,12] carriers, flame retardation, or as expansion agents to debond joints after a certain stimulus.

However, these added advantages always depend on the filler content and its compatibility with the adhesive, since a good bond between these two materials is necessary to attain proper filler performance [16]. In terms of filler content, two behaviors can be seen: a gradual growth of all particle amounts, typically seen for electrical [10] and thermal [11] capabilities; or a point of peak performance [10], usually seen in the mechanical performance of the joint that has an optimal filler value. This last behavior usually presents a “bell” shape, starting with an increase presenting a maximum benefit in the wanted property and then followed by a decrease that could go below the undoped state.

Several studies have been performed on these phenomena for epoxy resins; Ghabezi and Farahani [7] presented a critical review on the effect of nanoparticles, mostly ceramics, on the fracture toughness of composites. From the cases presented, most reached their peak performance for small weight percentages. In [8], the authors showed that 0.43% *w*/*w* of nano alumina showed the optimal performance for this particle on mode I and mode II for toughening glass fiber reinforced composite (GFRP). Or, in [15], 1% *w*/*w* of nano clay was the optimal value after having tested also higher amounts of 3% and 5%, that showed progressively similar performances for the neat state, respectively. Metallic fillers can also be typically used for electrical or thermal conductive applications. Darwish et al. [10] showed that in the case of metallic particles of different metals and sizes, their peak performance both in terms of strength and electrical conductivity was between 15% to 30% *w*/*w* depending on the particle used.

Nonetheless, even when reinforced, four main failure modes can be observed when analyzing broken adhesively joined specimens. They are represented in Figure 1. The most desired fracture mechanism is cohesive failure (a) in the adhesive, where the crack propagates along the adhesive, with the adherends’ surfaces still coated with a layer of adhesive. Such failure usually occurs when the adhesion between the adhesive and the adherend is very strong, and the substrates can support a higher load than the adhesive. Cohesive failure (b) in the adherend corresponds to the cases where the adhesive bond and the adhesive’s resistance are stronger than the adherend. The worst fracture pattern is the interfacial or adhesive failure (c), which results in the crack following a path along the interface between the adhesive and the adherends. This failure mode can appear as a consequence of incorrect surface preparation and, due to the low strength it provides, it is totally unacceptable in an industrial context. Lastly, alternating or mixed failure (d) involves the cases where the crack’s propagation path alternates between the adhesive and the adherends’ interfaces in the same failure process [1].

This said, another possible application of fillers is to change the failure mode of a bonded joint from the adhesive to cohesive failure. This is achieved by creating stress concentrations (SCs) inside the adhesive where the crack can progress, instead of maintaining itself close to the interface. Even though these fillers weaken the adhesive, the change from adhesive to cohesive is highly desired as it is more predictable, making their application worthwhile in cases where the presence of interfacial failure is inadmissible.

With respect to this subject and focusing particularly on the failure mode’s variation with the addition of glass microspheres, Hunter et al. [17] investigated how the presence of that type of particles influences the mechanical adhesion and behavior of single-lap joints with fiber-reinforced polymers for adhesives with different curing speed. Throughout this study, it was concluded that their use has different repercussions depending on the adhesive’s curing speed. The slow-curing adhesive showed a strength decrease when 3% *w*/*w* and 10% *w*/*w* concentrations of glass microspheres were added to the adhesive. On the other hand, the fast-curing epoxy adhesive revealed a joint strength increase for all doped configurations. Regarding the failure modes, thin-layer cohesive and fiber tear failure were the failure modes obtained for the slow-curing adhesive, while interfacial failure was also reported for the fast-curing adhesive, as well as the previously mentioned failure modes.

In a similar effort, Santos et al. [18] showed that the addition of 5% to 15% *v*/*v* of hollow glass beads to a 1K epoxy adhesive progressively changed the failure of mild steel adhesive single lap joint (SLJ) specimens from mixed, where half of the fracture surface was adhesive failure, to fully cohesive failure in the adhesive. Bruckner et al. [19] also researched the use of hollow glass microspheres on a 1K epoxy adhesive via SLJ specimens. The joints were tested in aged and unaged conditions. The results showed an improvement in the area of cohesive failure with a negligible effect on joint strength, similar to Santos et al. [18]. Aradhana et al. [20] used a combination of reduced graphite oxide (rGO) and silica hollow microspheres (SiHM) to produce a filler mixture that improved the conductivity of an epoxy adhesive. Results showed that the lap shear strength and unnotched impact strength were enhanced, whilst there was a decrease in tensile strength and notched impact strength. Nonetheless, both the thermal and electrical conductivity of the doped adhesive were improved substantially. This type of filler is currently being used in the formulation of commercial adhesives as reported by Ciardiello et al. [21] and Santos et al. [22]. The use of hollow glass microspheres has even been reported in the production of epoxy syntactic cellular foams [23], a lightweight composite that is intended to substitute the use of reticulated honeycomb and polyurethane expandable foams in the core of complex-shaped sandwich composite structures.

However, this procedure becomes quite expensive and time consuming; when predicting a change in failure mode, the use of experimental tests with several percentages of these fillers is needed. To enable this in a simpler and faster way, numerical models can be used, where the failure modes of joints bonded using adhesives doped with fillers can be predicted. Several techniques can be used to simulate adhesive joints, and the cohesive zone model (CZM) is widely implemented. The basis of CZM is the use of a cohesive material that simulates damage initiation and propagation. This material model was first proposed by Alfano and Crisfield [24]. It is capable of predicting the fracture behavior, the damage initiation and propagation, and the joint’s strength by using traction separation laws that dictate the behavior of its cohesive elements [25]. This method is able to model non-linear processes while avoiding stress singularities, as well as not needing the existence of a pre-crack nor the need for user interference for its propagation to occur [26]. However, using CZM requires defining the path where the crack will occur beforehand, limiting its ability to predict how a failure process will take place, making it non applicable for this purpose.

The extended finite element method (XFEM), on the other hand, can be used to predict the joint’s failure path as it allows the crack to grow freely [27,28] without the need for a precrack or a known crack path. This method is an extension of the finite element method (FEM), as the name implies. In fact, XFEM was developed since the FEM, which allows us to precisely predict the stresses and other mechanical phenomena acting on bonded structures [29,30], showed difficulties in modelling discontinuities such as cracks’ propagation.

The XFEM was created at the end of the 20th century as an extension of the traditional finite elements’ method, having been first introduced by Belyschko and Black in 1999. This method allows the crack to be modeled independently of the mesh without being dependent of any kind of remeshing method, modelling its path by itself [31]. It can be defined as a minimal remeshing finite element method and is characterized for introducing discontinuous enrichment functions to the conventional finite element displacement function in order to properly simulate the presence of the crack [27,32]. When there is damage propagation, phantom nodes are formed to subdivide the elements that are intersected by the crack and enable the existence of discontinuities between the new elements. This allows the crack to propagate along an arbitrary path [31].

Regarding the application of the XFEM in the numerical study of adhesively bonded joints, da Silva et al. [33] used this method to predict the failure mode of single lap joints reinforced with cork microparticles. This study showed that XFEM can be used as a valid method to predict the failure pattern as well as the mechanical performance of bonded joints. The failure modes that were experimentally obtained were fully validated numerically, and it has been observed that by increasing the size and the amount of the particles, the crack evolves from an adhesive failure to a cohesive one.

Campilho et al. [28] also studied the applicability of the XFEM to simulate the mechanical behavior of DCB specimens, as well as their crack propagation process. The numerical results were compared to the experimental *P*-*δ* curves and the critical strain energy in the pure mode I values. The comparison between these results proved that the XFEM provides a reliable simulation of the behavior of adhesively bonded DCB specimens and can therefore be used to correctly model bonded structures.

This study was developed as a continuation of two previous experimental investigations in partnership with ArcelorMittal Global R&D (Montataire, France). Firstly, Santos et al. worked on experimentally changing the failure mechanisms of bonded SLJ from mixed failure to fully cohesive by adding hollow glass beads (GBs) to the adhesive mixture [18]. Then, they investigated the effect of these inorganic fillers on the tensile strength and mode I fracture properties of the studied adhesives [22].

From all this experimental work to evaluate the influence of 0, 5, 10 and 15% of particles on two different adhesives, several relevant conclusions were drawn, and the purpose of the project was met, but only after an extensive number of tests were performed. As such, in this work, a numerical model intent on reproducing the change in failure mode seen in experimental data by means of filler addition was performed in ABAQUS© 2017b (Dassault Systems, Providence, RI, USA) to reduce the number of tests needed to achieve the same conclusions.

This model was devised in keeping with the XFEM, simulating not only the presence of the adhesive, but also introducing the glass beads into the model, reproducing their interactions during the joint’s failure. This ran contrary to the common method that uses the cohesive properties of the doped adhesive to devise a CZM formulation requiring experimental tests for each configuration. Additionally, this approach cannot simulate the filler-adhesive failure mechanics, as CZM always assumes cohesive failure.

The numerical model was reproduced for three conditions: one considering the neat adhesive; another with 10% *v*/*v* of hollow glass beads introduced into the adhesive layer, being this is the best compromise between strength and intended fracture mode [18]; and finally, an intermediate value to help validate this model, 5% of hollow glass beads. This simulation allowed us to assess not only the crack’s path and the corresponding failure mode, but also the strength of the bonded joints. The numerical results were compared to the ones previously obtained experimentally.

## 2. Project Contextualization

Being that this is a numerical extension of the previous experimental investigation on SLJ failure mechanisms [18], this section introduces the discoveries of the previous research to contextualize the results of this paper. These experimental SLJ specimens were manufactured and tested following the recommendations of [34], in accordance with standards ISO 4587:2003 [35] and ASTM D1002-10(2019) [36].

The experimental *P*-*δ* curves obtained for the SLJs with mild steel are presented in Figure 2 for quasi-static conditions. These curves were compared against the experimental counterparts.

From these representative results, a few clear trends can be seen: the joint’s stiffness is similar in all cases, each configuration shows clear plastic deformation of the substrates, and the loss of maximum extension with increasing content of filler particles. To account for all the SLJ tests, the failure loads and displacements at failure of all configurations are represented in Figure 3a,b.

As one can conclude by looking at the experimental results, the mechanical performance of the bonded joints tends to decrease with an increasing volume of glass particles. However, even though the failure load values show a global reduction with the rising number of glass beads, the differences are not especially significant (−3%) until 10% *v*/*v* of glass beads. However, considering the 15% configuration not shown here, this variation can go up to −7% [18].

Nonetheless, an analysis of the maximum displacement values as a function of the added % *v*/*v* of GBs shows a significant decrease of about −20% for the 10% configuration. This suggests that the adhesive mixture not only loses ductility, but also its energy absorption capacity.

Another important aspect to analyze in this work is the failure mode, which is the main purpose of the addition of these particles to the adhesive mixture. As such, Figure 4 illustrates the fracture surfaces of the SLJs for each amount of GBs.

From the failed joints it is possible to observe that the addition of GBs changed the fracture surfaces. More specifically the decrease in percentage of area of adhesive failure, mostly localized in the ends of the bonded joints. As such one can infer that, by addition of more glass beads, the failure mode transitions from adhesive to cohesive failure. This phenomenon stabilized around 10% to 15% *v*/*v* of GBs [18], therefore the numerical simulations were solely performed until the 10% configuration.

This unwanted adhesive failure mechanism is associated with the high stress concentration in the corners of the joint plus the balance between cohesive and adhesive strength. For the neat adhesive the cohesive strength is prevalent resulting in crack onset at the multi-material corners and subsequent propagation through the interface.

However, by adding particles new SCs are introduced in the material, now inside the adhesive due to the local stress fields around the GBs. This shifts the crack path from the interface towards the middle of the adhesive, having an earlier onset the more GBs were introduced.

## 3. Numerical Details

In this section, the numerical procedures used to simulate SLJs with an increasing content of GBs are presented, from neat adhesive to 5 and 10% *v*/*v* of GBs. Being that the main goal of this work is to comprehend the influence of the glass beads on the crack’s propagation path of the SLJ, the numerical results were compared against the previously determined experimental data [18].

The specimen’s failure mode, which should evolve from adhesive to cohesive as experimentally evidenced [18], was the main topic analyzed. However, the numerical *P*-*δ* curves were also collected and compared against experimental results to better understand the impact of the GBs on the joint’s performance.

### 3.1. Model Parameters

The selection of the numerical model configuration, element types used in the mesh and other numerical parameters will strongly influence the result’s accuracy, requiring a careful definition of the model parameters to ensure accuracy.

Due to the non-linear character of the simulation at hand and to simplify the study, a 2D planar analysis was performed. To obtain a smooth crack propagation, auto adjustable time increments with a maximum increment of 0.5% of the applied displacements were used.

It is also relevant to state that the model does not consider the presence of defects in adhesive layers nor in the interfaces between the adhesive and the substrates.

### 3.2. SLJ Geometry

The SLJ’s geometry adopted in the model is the same as the one that was tested in the experimental procedure (Figure 5).

Nonetheless, numerically, there is the need to define the position of the GBs introduced in the adhesive. As such, to create a proper SLJ model, a MATLAB© 2021 (Mathworks, MA, USA) script was developed, which randomly plots the intended glass spheres along the adhesive layer. The coordinates generated by this script were introduced in AutoCAD© 2021, and the model geometry was imported into Abaqus© 2017b.

### 3.3. Materials

In this work, an epoxy adhesive doped with hollow glass beads was used to produce mild steel SLJs. The general properties and characteristics of these materials were described in the next sub-sections. The mechanical properties used in the simulation were detailed later.

To simplify the relation between this work and previous papers constantly cited in this work wherein the materials used were provided by the same partner company, the previously published nomenclatures were kept for the adhesive [18,22,37,38], “A”, the fillers [18,22], “GBs”, and the substrate material [18], “mild steel”.

#### 3.3.1. Adhesive

Adhesive A is a one-component epoxy and crash-resistant adhesive supplied by the partner company, ArcelorMittal Global R&D. The mechanical properties of Adhesive A were determined in previous works, first by Borges et al. [37,38]; however, more recently and for this project, the tensile and mode I properties were retested by Santos et al. [22]. For tensile loading [22], bulk specimens were used; thick adherend shear test (TAST) specimens characterized the shear properties [24]; for mode I fracture [22], double cantilever beam (DCB) tests were performed; and finally, end-notched flexure (ENF) specimens were sued for the mode II fracture toughness [38]. These were carried out to determine the tensile strength and Young’s modulus (bulk specimens), *σ_f_* and *E*, respectively; the shear strength (TAST specimens) is *τ_f_*; the fracture toughness in mode I (DCB specimens) is *G*_IC_; and the fracture toughness in mode II (ENF specimens) is *G*_IIC_.

#### 3.3.2. Fillers

The GBs are thin-walled hollow spheres made of soda–lime–borosilicate glass, referred to by the manufacturer (3M—St. Paul, MN, USA) as K37 glass bubbles [39]. Their size and density can vary, and they are characterized by their lightweight, high strength-to-density ratio, and a considerable isostatic crush strength [39,40].

Table 1 lists the main properties of the hollow glass beads used to dope the studied adhesives.

#### 3.3.3. Substrates

The substrates used in this work were manufactured from mild steel sheets, supplied by the partner company, ArcelorMittal Global R&D.

### 3.4. XFEM Formulation

Although the XFEM methodology implemented in Abaqus© allows us to define a precrack for damage initiation, the models developed for this study did not employ this feature. This was carried out in order to allow the crack to freely propagate along the adhesive layer.

#### 3.4.1. Formulation

In Figure 6, one can observe two representative images of the propagation of a crack in the XFEM elements. When a crack propagates through an element, it gets partitioned into two sub-elements, Ω*_A_* and Ω*_B_*. Initially, the element represented in (a) had nodes *n*_1_ to *n*_4_, and after being intersected by the crack, four phantom nodes, ñ_1_ to ñ_4_, were created to allow the existence of discontinuities between the new sub-elements. Therefore, the element represented in (a) was replaced by two new sub-elements in (b): Ω*_A_*, composed of the nodes *n*_1_, *n*_2_, ñ_3_ and ñ_4_; and Ω*_B_* composed of the nodes ñ_1_, ñ_2_, *n*_3_ and *n*_4_ [31].

Equation (1), presented below, represents the displacement vector associated with this method [31].
(1)$$u=\sum_{I=1}^{N}\;N_{I}\left(x\right)\left[u_{I}+H\left(x\right)\;a_{l}+\sum_{\alpha =1}^{4}F_{\alpha }\left(x\right)\;b_{I}^{\alpha }\;\right]$$
where *N_I_*(*x*) represents the nodal shape function, and u_I_ is the nodal displacement vector of the continuous part of the formulation [31]. Moreover, *H*(*x*)a_l_ is the generalized Heaviside enrichment function, and *F_α_*(*x*)$b_{I}^{\alpha }$ accounts for the nodes whose shape function is intersected by the crack.

The term *H*(*x*) (Equation (2)) can be generally designated as the jump function or discontinuous shape function, which allows us to model the discontinuities across the crack over the points along the crack surface [27].
(2)$$H(x)=\left\{\begin{array}{cc} 1, & \mathrm{if}\;(x-x^{*})\;\cdot n\;\geq 0\\ -1, & \mathrm{otherwise} \end{array}\right.$$
being that *x* is a sample Gauss integration point, *x*^∗^ is the point of the crack closest to *x*, and *n* is the unit vector normal to the crack at *x*^∗^ [41].

The term *n* is multiplied by a_l_, being the nodal enriched degree of freedom vector [31]. Another term that also stands for the nodal enriched degree of freedom vector is $b_{I}^{\alpha }$. Moreover, *F_α_*(*x*) represents the asymptotic crack-tip functions [31].

The XFEM can generally follow two damage-modelling approaches: the cohesive segments approach, and the Linear Elastic Fracture Mechanics (LEFM) approach [31].

The cohesive segment approach is the most widely used for the modelling of bonded joints. This approach is governed by traction–separation laws, and the damage properties are defined as part of the bulk material definition.

Three stress-based and three strain-based damage initiation criteria can be used while following this method. And the crack’s initiation depends on the stress/strain value at the center of the enriched elements [31]. The Macaulay Brackets are used in order to specify that a purely compressive stress does not induce damage [42].

The maximum principal stress (MAXPS), Equation (3), and maximum principal strain (MAXPE), Equation (4), criteria:

(3)$$f=\frac{\left\langle \sigma_{n}\right\rangle }{\sigma_{max}^{0}}$$
where *σ_n_* represents the maximum principal stress at an integration point, and $\sigma_{max}^{0}$ is related to the material strength in terms of tension.
(4)$$f=\frac{\left\langle \varepsilon_{n}\right\rangle }{\varepsilon_{max}^{0}}$$
where *ε_n_* is the maximum principal strain at an integration point, and $\varepsilon_{max}^{0}$ is the material strength in terms of deformation [42].

For these criteria, the crack initiates when *f* = 1, and its plane is perpendicular to the direction of the maximum principal stress [31].

The maximum nominal stress (MAXS), Equation (5), and maximum nominal strain (MAXE), Equation (6), criteria:

(5)$$f=MAX\left\{\frac{\left\langle \sigma_{n}\right\rangle }{N_{max}},\frac{\sigma_{t}}{T_{max}},\frac{\sigma_{s}}{S_{max}}\right\}\;\;\;\;\left\langle \sigma_{n}\right\rangle =\left\{\begin{array}{c} {\;\sigma }_{n}\;,\;\;\;\mathrm{if}\;\sigma_{n}>0\;\;\\ \;0\;,\;\;\;\mathrm{if}\;\sigma_{n}<0 \end{array}\right.$$
where *σ_n_* and *N_max_* are the maximum nominal stress and the critical stress values in a normal-only mode at an integration point; *σ_t_* and *T_max_* are the maximum nominal stress and the critical stress values in a shear-only mode along the first shear direction; and *σ_s_* and *S_max_* are the maximum nominal stress and the critical stress values in a shear-only mode along the second shear direction [31].
(6)$$f=MAX\left\{\frac{\left\langle \varepsilon_{n}\right\rangle }{\varepsilon_{n}^{max}},\frac{\varepsilon_{t}}{\varepsilon_{t}^{max}},\frac{\varepsilon_{s}}{\varepsilon_{s}^{max}}\right\}\;\;\left\langle \varepsilon_{n}\right\rangle =\left\{\begin{array}{c} {\;\varepsilon }_{n}\;,\;\;\;\mathrm{if}\;\varepsilon_{n}>0\;\;\\ \;0\;,\;\;\;\mathrm{if}\;\varepsilon_{n}<0 \end{array}\right.$$
where *ε_n_* and $\varepsilon_{n}^{max}$ are the maximum nominal strain and the critical strain values in a normal-only mode at an integration point; *ε_t_* and ${\;\varepsilon }_{t}^{max}$ are the maximum nominal strain and the critical strain values in a shear-only mode along the first shear direction; and *ε_s_* and $\varepsilon_{s}^{max}$ are the maximum nominal strain and the critical strain values in a shear-only mode along the second shear direction [31].

For these cases, the damage initiates when 1.0 ≤ *f* ≤ 1.0 + *f*_tol_, being *f_tol_* a tolerance value specified by the user, and the user may also define the crack plane normal direction [31].

The quadratic nominal stress (QUADS), Equation (7), and quadratic nominal strain (QUADE), Equation (8), criteria:


(7)
$${\left(\frac{\left\langle \sigma_{n}\right\rangle }{N_{max}}\right)}^{2}+{\left(\frac{\sigma_{t}}{T_{max}}\right)}^{2}+{\left(\frac{\sigma_{s}}{S_{max}}\right)}^{2}=1$$



(8)
$${\left(\frac{\left\langle \varepsilon_{n}\right\rangle }{\varepsilon_{n}^{max}}\right)}^{2}+{\left(\frac{\varepsilon_{s}}{\varepsilon_{s}^{max}}\right)}^{2}+{\left(\frac{\varepsilon_{t}}{\varepsilon_{t}^{max}}\right)}^{2}=1$$


For these criteria, the user also defines the crack plane as normal, as for MAXS and MAXE [31].

#### 3.4.2. Material Properties

Regarding the properties that were attributed to each material, an elasto-plastic isotropic response was considered for the mild steel adherends. This kind of response was chosen due to the fact that, during the experimental tests, considerable plastic deformation of the substrates was encountered. The elastic properties of the steel are presented in Table 2.

The adhesive mixture was modeled recurring to a triangular traction–separation law, similar to the one that is portrayed in Figure 7, which assumes a linear softening.

This type of law is defined by three properties: the *σ_i_*, *K_i_* and *G_i_*_C_, which represent the cohesive strength, stiffness, and toughness, respectively. Each loading mode, tension (I) and shear (II), has its own cohesive law. The triangular traction–separation law considers an initial linear elastic behavior until *σ_i_* is reached. Then, a linear evolution until damage follows, which is associated with the *G_i_*_C_, i.e., the area below the law [40].

The crack initiation criterion used was based on the quadratic stress criterion (QUADS). But since each material, adhesive and GBs was simulated as a separate entity, each one has its specific CZM laws. As such, the properties assigned for the damage law (Figure 7) associated with this criterion were the ones represented in Table 2 for the adhesive and the glass beads. Each material, adhesive and GBs, was simulated as separate entities; therefore, each one has its specific CZM laws.

The standard deviations of the properties of the adhesive can be seen in each of the characterization works described previously, since only the mean values were used as inputs. The mechanical properties of the glass beads introduced into the numerical model were determined indirectly via a reverse analysis. In Table 2 only the elastic mechanical properties of the material are presented, nonetheless, the simulation used the yield curve of the mild steel to account for material plasticity.

### 3.5. Mesh

The model was meshed with 4-node bilinear plane strain elements (CPE4R), with reduced integration. The meshes were created with a single bias effect from the substrate’s free edges towards the adhesive bond, as well as a double bias from the center towards the edges of the bondline. This was conducted since the joint’s bimaterial corners are expected to exhibit larger stress gradients (stress concentrations—SCs), which were found to require finer meshes to be correctly analyzed.

As one can observe in Figure 8, which presents the representative meshes of the SLJs models created, the mesh was further refined for the cases of doped SLJs. These configurations call for smaller element sizes in order to properly mesh their geometries. The region around the glass beads were also further refined, minimizing large element size transitions.

### 3.6. Boundary Conditions

The boundary conditions established during these simulations were the ones that are presented in Figure 9, which simulate the loads applied during real testing (Figure 10).

The restraining condition, on the left side of the joint, blocked the horizontal and vertical movement, as well as the rotation in the clamped end of the specimen. The loading condition implied a horizontal displacement (*δ_x_*), while the vertical movement and rotation were restricted.

### 3.7. Simulation Outputs

The load and displacement of each joint were recorded in the simulation and compared against the experimental measurements of the test machine (Figure 10).

Additionally, the vonMises stress distributions and crack propagation paths were extracted in the numerical model. This information was compared against experimental data obtained using the camera setup (Figure 10); the deformation of the substrates and bondline were recorded, as well as the post-failure images (Figure 4).

## 4. Results and Discussion

In this section, the results of this numerical work are presented and compared against previous experimental data. Different SLJ configurations previously tested were simulated in order to reproduce and study their complex failure phenomena. Two conditions were evaluated through the crack path evolution, the *P*-*δ* curves and the failure mechanism, presented in this order.

The numerical *P*-*δ* curves obtained for the SLJs with mild steel are presented in Figure 11 for Adhesive A at quasi-static conditions. These curves were compared against the experimental counterparts.

As one can observe in Figure 11, very significant plastic deformation of the substrates is taking place, since the *P*-*δ* curves show a transition from an elastic to a plastic behavior after reaching a load value around 3000N. Being that the substrate material is a thin mild steel, it is natural that plastic yielding is ruling the SLJ’s behavior. Comparing the experimental and the numerical results, it becomes possible to conclude that the shift from the elastic to the plastic behavior is more abrupt in the numerical model. This can be explained by the fact that the input stress–strain curve of the substrate is composed of discrete points extracted from the mild steel’s tensile curve in order not to minimize the computational times associated with the numerical study.

Regarding the moment of failure, the maximum loads and displacements obtained both experimentally and numerically are presented in Figure 12a,b, respectively.

The relative errors between the experimental and numerical values are detailed in Table 3.

Regarding these results, the numerical studies predicted slightly higher failure loads and displacements for all configurations, as one can confirm in Figure 12 and Table 3. These differences were higher for the neat adhesive, which showed 8.8% and 20.6% variation in terms of load and displacement, respectively. The doped configurations presented smaller variations of about 2% and 10% for each respective variable.

However, and considering the fact that numerical simulations assume the existence of perfect conditions throughout the test, these differences should be considered as expectable. Moreover, and by globally analyzing the evolution of the adhesive’s mechanical performance, the performance degradation caused by the presence of the GBs on the adhesive’s layer is captured by the numerical models. This is proved not only by the maximum displacement, but also the area below the *P*-*δ* curves that gradually decreases with the increasing number of particles.

Figure 13 compares representative images of the plastic deformation of the substrates during testing for the experimental and the numerical results. The comparison between the two images allows us to confirm that the deformed shape obtained through the numerical simulation closely resembles the phenomenon that was detected experimentally.

The change in failure mechanism of the SLJ specimens through particle addition can be clearly seen in Figure 14, which illustrates the crack’s propagation path of each configuration.

The stiffness degradation parameter (SDEG) was used to predict such a path since it represents the level of damage of the elements. A value of zero corresponds to an undamaged element (in blue), and an element is considered to be fully damaged (in red) whenever this parameter reaches a value of one. The numerical results show that by introducing GBs into the adhesive layer, the crack initiation path moves from the interface to the middle of the adhesive. This results in the failure mode evolving from adhesive to cohesive. Such a phenomenon is in accordance with what was reported experimentally (Figure 4), where the influence of the glass microspheres clearly changed the crack’s propagation path.

Specifically focusing on the stress distribution of a neat and a doped configuration, as seen in Figure 15, the stress concentrations (SCs) present in the adhesive layer can be easily identified.

In the neat state, both experimentally (Figure 4) and numerically (Figure 14), the crack path undoubtedly begins propagating through the interfacial region. This occurs since the most critical stress region (in the adhesive) is located in the multi-material corners (SC_1_) at the ends of the bonded part of the joint.

However, with regards to the doped SLJs, an initial short interfacial crack can be observed experimentally (Figure 4). Numerically, this phenomenon does not occur, since presence of particles within the adhesive layer accumulates more critical stress fields in their periphery than those near the ends of the bonded joints, as one can confirm on the right scheme of Figure 15.

As such, we can conclude that even though a sharp corner stress concentration is still present in the joints with added particles, being hollow, the glass microspheres (SC_3_) have a highly damaging effect. Furthermore, as seen in Figure 15, their distribution near SC_2_ makes the stress concentration field go further into the middle of the adhesive. At failure, the crack forms by linking SC_2_ to SC_3_ through the closest path (middle of the adhesive), resulting in the paths seen in Figure 14.

In more detail, Figure 16 presents a closer look at the crack’s propagation path in SLJ with the addition of 10% volume of glass beads. In this figure, the location of the glass particles is highlighted (in green), and the crack path is represented (in red), enabling us to see that the glass beads’ presence clearly influences the crack’s path. As one can observe, the crack propagates through the regions where the glass beads are present, since these create stress concentrations along its periphery. Considering that these represent weaker points in the adhesive matrix since their local damage level is significantly higher, they stipulate the crack propagation path.

However, experimental results showed that the crack always starts propagating in the interfacial region. This proved true even for the doped configurations, but with a much smaller or even negligible adhesive failure area, as seen in Figure 4.

This phenomenon could not be replicated in the simulations due to the fact that the experimentally tested SLJs contain additional factors that are not considered in this simulation. From imperfectly shaped corners to high peel stresses originating from the plastic yielding of the substrates, all these factors accumulate significant stress fields at the ends of the bonded joint, which force the crack to initially propagate interfacially for a short period. Since the factors described above are not accounted for in the development of the numerical models, these might explain the slight differences between the experimental and the numerical results, with regards to the initial part of the crack’s propagation path.

## 5. Conclusions

In this work, adhesively bonded single lap joints with mild steel substrates were numerically tested. These joints were previously studied experimentally, where, in the neat state, a severe adhesive failure issue was identified. To solve this problem, several adhesive configurations, doped with different amounts of hollow glass beads, were experimentally studied and compared with regards to the joint’s strength and change in failure mode [18].

To prevent future extensive experimental work to study different conditions in similar situations, the same joint configurations were numerically modeled in order to analyze the evolution of the failure pattern and the impact on the joint’s strength and ductility. The main conclusions extracted from this study are the following:The introduction of hollow glass beads on the adhesive layer has a negative influence on its mechanical performance as it weakens the adhesive by decreasing the joint’s maximum extension, as well as the adhesive’s ability to absorb energy.It was observed that adding glass particles to the adhesive clearly influences the joint’s failure mode. In fact, their presence creates local stress fields around their periphery, which forces the crack to propagate through those regions. Therefore, one can conclude that adding this type of particle allows the joint to have a cohesive failure rather than an adhesive one.The XFEM was found to be an effective method for predicting the failure modes of the bonded joints since the numerical results showed that adding glass particles to the adhesive layer results in a cohesive failure. Moreover, it provided satisfactory estimations of the SLJs failure loads, as well as its maximum extension values. Finally, it is also able to accurately reproduce the experimentally observed plastic yielding of the substrates.

To improve this work, the implementation of a different cohesive law for the adhesive, using for example a trapezoidal law, could improve the similarity between experimental and numerical *P*-*δ* curves. Additionally, validating this prediction tool for higher GB content or other adhesives doped with these particles could further solidify the validity of this numerical prediction model.

## Figures and Tables

**Figure 1 materials-16-07499-f001:**
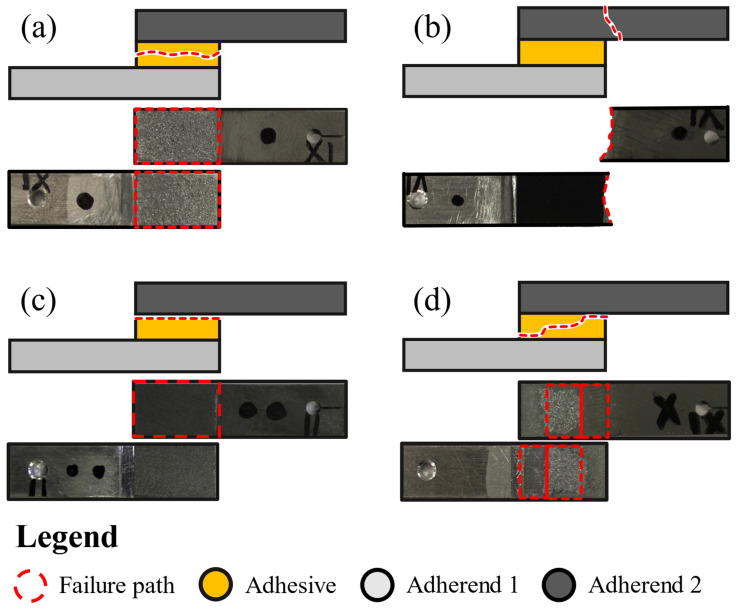
Schematic representation and experimental examples of the different types of joint failure. Experimental images related with the work of [12] using small-scale SLJs (unpublished images). (**a**) Cohesive failure in the adhesive, (**b**) cohesive failure in the adherend, (**c**) interfacial or adhesive failure, and (**d**) alternating or mixed failure. Failure path in red dashed line, adhesive in yellow, bottom adherend in light grey and top adherend in dark grey.

**Figure 2 materials-16-07499-f002:**
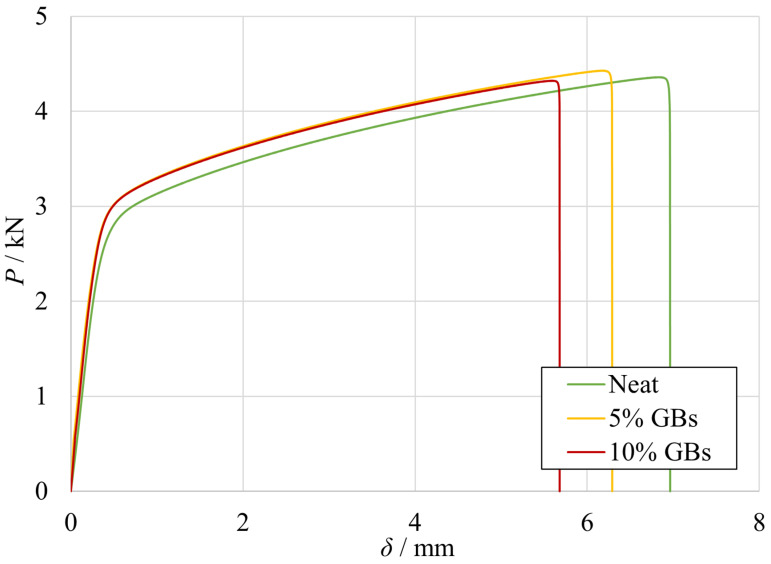
Representative experimental *P*-*δ* curves of SLJs with different adhesive configurations: neat (green), 5% (yellow) and 10% (red) *v*/*v* of GBs. Adapted from [18].

**Figure 3 materials-16-07499-f003:**
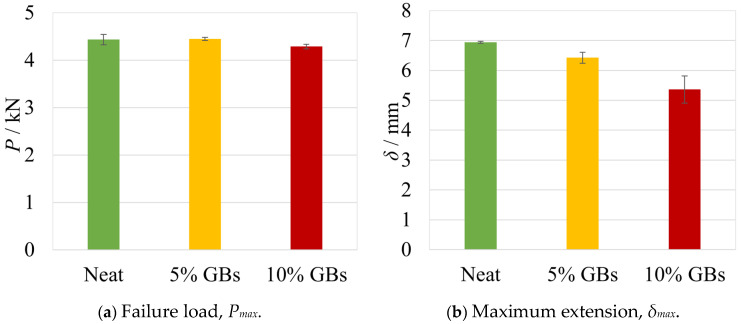
The failure load and maximum extension values for the different percent volumes of hollow glass beads. Adapted from [18].

**Figure 4 materials-16-07499-f004:**
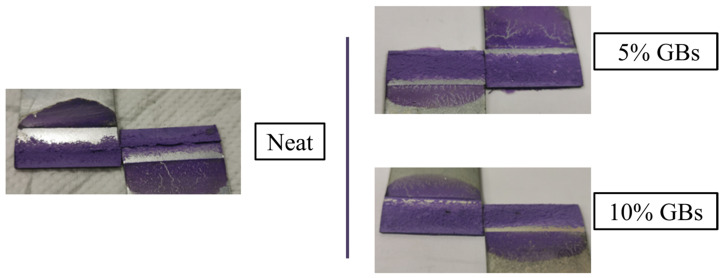
SLJs fracture surfaces as a function of the % *v*/*v* of GBs added to the adhesive mixture. Adapted from [18].

**Figure 5 materials-16-07499-f005:**
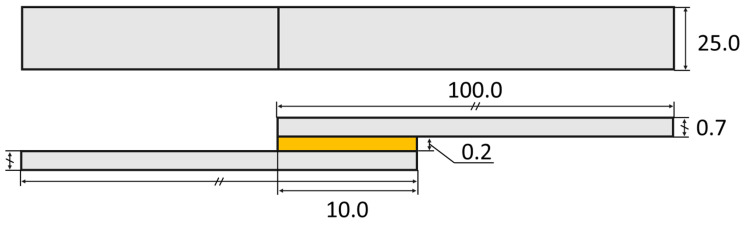
Schematic representation of the SLJ specimens’ geometry, dimensions in millimeters. Adherend in grey and adhesive in yellow.

**Figure 6 materials-16-07499-f006:**
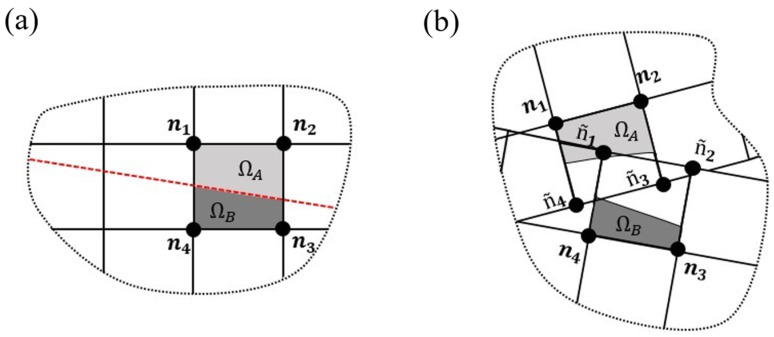
Representative scheme of the partition phenomenon associated with the propagation of a crack in XFEM. Crack path presented in a red dashed line.

**Figure 7 materials-16-07499-f007:**
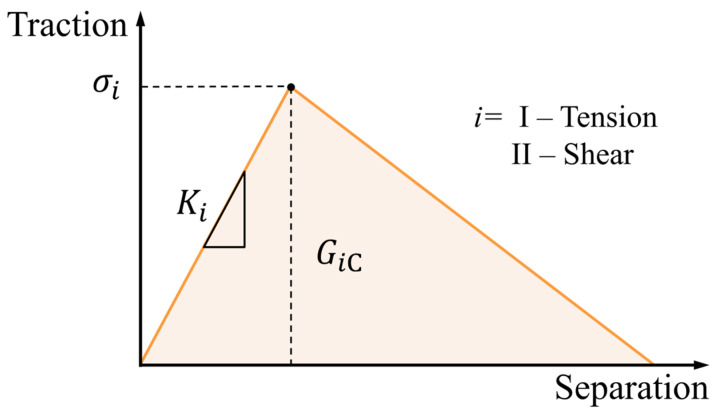
Schematic representation of the triangular traction separation law. *σ_i_*, *K_i_* and *G_i_*_C_ are the maximum cohesive strength, stiffness, and fracture toughness of the element, respectively.

**Figure 8 materials-16-07499-f008:**
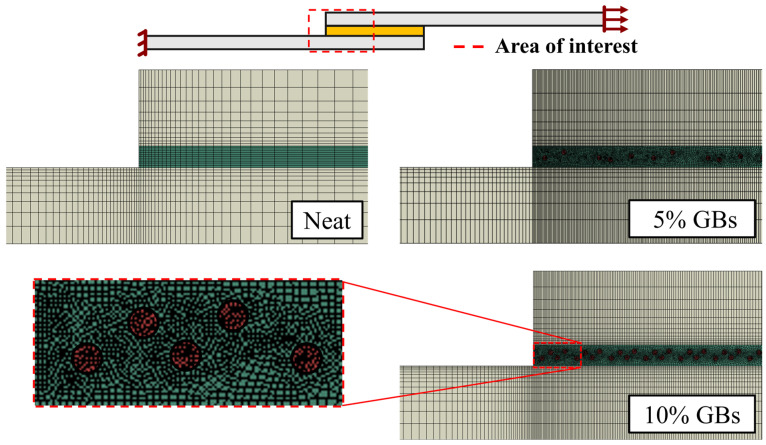
Representative meshes of the SLJs models with neat adhesive, addition of 5% *v*/*v* GBs and 10% *v*/*v* GBs. Adherend in grey, adhesive in green, and spheres in red.

**Figure 9 materials-16-07499-f009:**
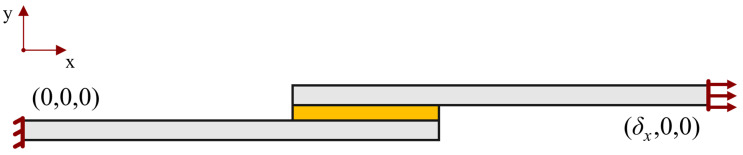
Boundary conditions established for the simulation of the SLJ behavior.

**Figure 10 materials-16-07499-f010:**
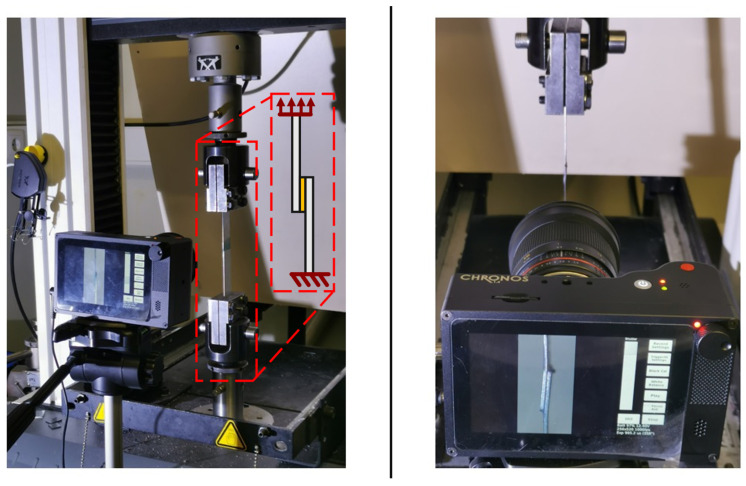
Testing setup used to analyze the SLJ specimens in the prior experimental work [18]. Schematic comparison (**left**, highlighted in red dashed line) between the numerical model (Figure 9) and the experimental setup. Bottom fixture is the fixed boundary condition, and top fixture is the moving head (*δ_x_*) of the machine. High speed camara (**right**) was used to observe the failure of the joints, later compared against the numerical simulations.

**Figure 11 materials-16-07499-f011:**
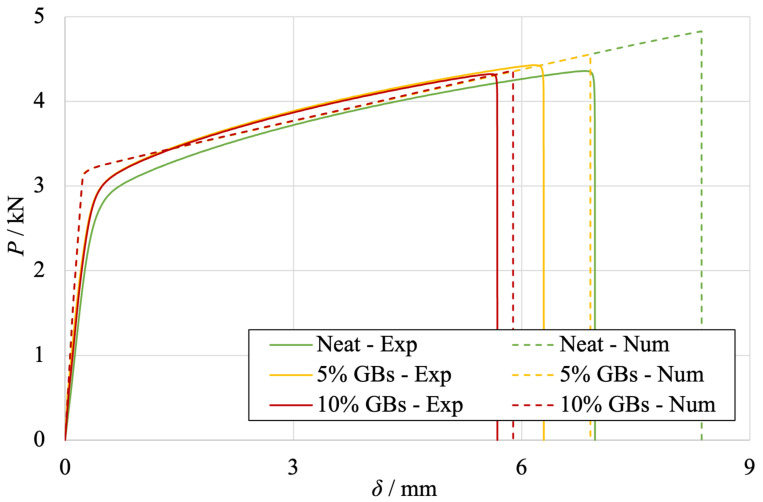
Comparison between the experimental (full line) and numerical (dashed line) *P*-*δ* curves of SLJs with different adhesive configurations: neat (green), 5% (yellow) and 10% (red) *v*/*v* of GBs. Experimental data from Figure 2.

**Figure 12 materials-16-07499-f012:**
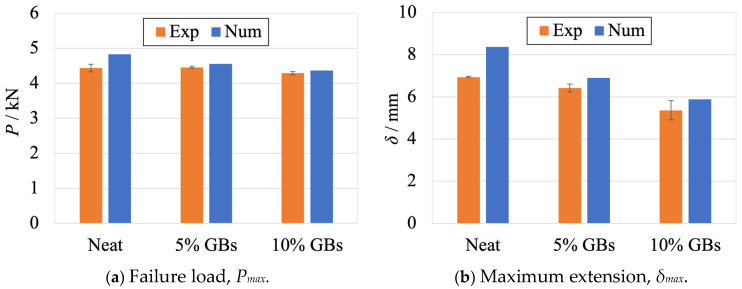
Comparison between the experimental (orange) and numerical (blue) results at failure of SLJs with different adhesive configurations: neat, 5% and 10% volume of GBs. Experimental data from Figure 3.

**Figure 13 materials-16-07499-f013:**
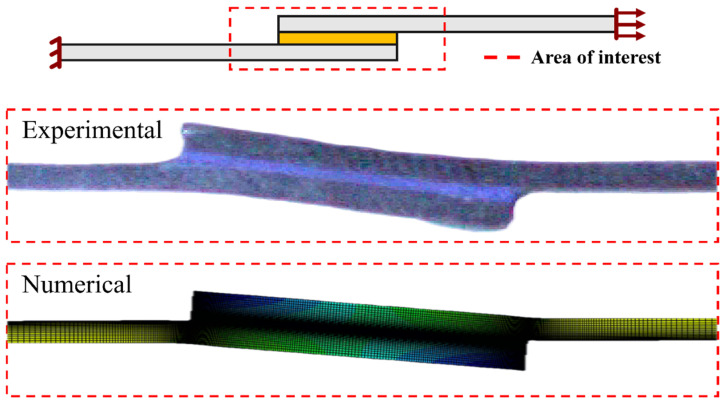
Representative images of the plastic deformations to which the mild steel substrates are subjected, experimental (**top**) and numerical (**bottom**) results.

**Figure 14 materials-16-07499-f014:**
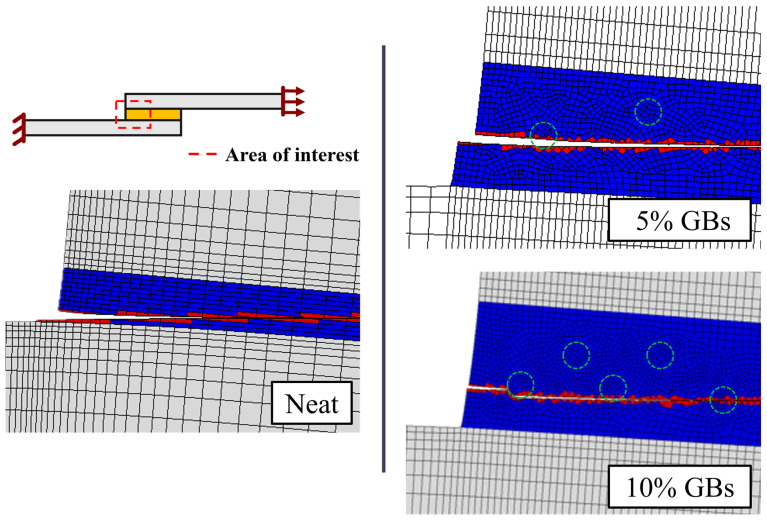
SLJs failure modes with neat adhesive, 5% and 10% *v*/*v* of GBs. Red and blue areas represent the fully damaged and undamaged adhesive, respectively. Grey elements represent the substrates, and green dashed circles identify the position of the GBs in the adhesive.

**Figure 15 materials-16-07499-f015:**
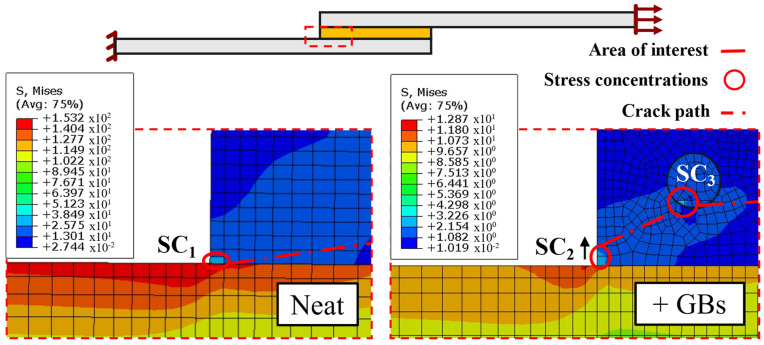
Representative local stress fields on the SLJs with neat adhesive (**left**), and with addition of particles (**right**), obtained from the numerical results at the same applied displacement. Stress concentrations (SCs) are numbered simply for referencing purposes.

**Figure 16 materials-16-07499-f016:**
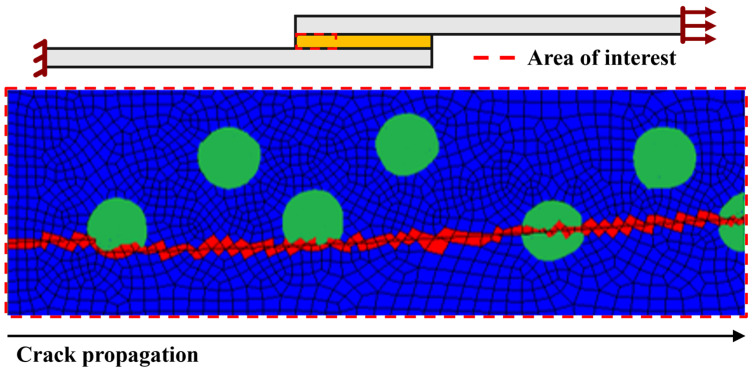
SLJ failure mode with the addition of 10% volume of glass beads to the adhesive. Red and blue areas represent the fully damaged and undamaged adhesive, respectively. Green represents the added particles.

**Table 1 materials-16-07499-t001:** Properties of the GBs presented by the supplier’s datasheet [39].

Property	GBs
*ρ/*gcm^−3^	0.37
$\bar{d}$/µm	45
*σ_c_*/MPa	20.6

**Table 2 materials-16-07499-t002:** Material properties used in the numerical study.

Property	Adhesive A	GBs	Mild Steel
*K_I_*/MPa	2137 ^1^	3500	210,000
*ν*	0.30	0.23	0.33
*K_II_*/MPa	822	1423	78,947
*σ_I_*/MPa	30.2 ^1^	1.0	−
*σ_II_*/MPa	30.9 ^2^	0.8	−
*G*_IC_/Nmm^−1^	2.60 ^1^	0.25	−
*G*_IIC_/Nmm^−1^	10.70 ^2^	0.10	−

Obtained: ^1^ [22]; ^2^ [38].

**Table 3 materials-16-07499-t003:** Relative errors between the experimental and numerical results at failure of SLJs with different adhesive configurations: neat, 5% and 10% volume of GBs.

Configurations	∆*P_max_*/%	∆*δ_max_*/%
Neat	8.8	20.6
5% GBs	2.3	7.5
10% GBs	1.7	9.8

## Data Availability

The data presented in this study are available on request from the corresponding author. The data are not publicly available due to a confidentiality agreement with ArcelorMittal Global R&D, Montataire, France.

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
