# Peer review of "Extended Finite Element Method (XFEM) Model for the Damage Mechanisms Present in Joints Bonded Using Adhesives Doped with Inorganic Fillers"

_materials, 2023, doi:10.3390/ma16237499_

Round 1
Reviewer 1 Report
Comments and Suggestions for Authors
Addressing the inefficiency in studying adhesive interface failure modes through experimental techniques, the application of eXtended Finite Element Method (XFEM) significantly accelerates the research process. This study, grounded in the XFEM approach, utilizes ABAQUS software to explore numerical models under diverse conditions, encompassing neat adhesive, 5% GBs, 10% GBs, and 15% GBs, while comparing these models with experimental data. The research content is comprehensive and logically organized. Furthermore, the references are aptly cited. Nevertheless, the article appears to present a superficial application of established XFEM technology, lacking a focus on crucial scientific issues within the adhesive field, and thereby failing to demonstrate innovation. Moreover, the poor clarity of Figures 1, 3, 14, 15, and 16 seemingly reflects a lack of meticulousness in the author's scientific approach. As a result, it is recommended to reject this article.
Comments on the Quality of English LanguageThe language expression is very good.
Author Response
Addressing the inefficiency in studying adhesive interface failure modes through experimental techniques, the application of eXtended Finite Element Method (XFEM) significantly accelerates the research process. This study, grounded in the XFEM approach, utilizes ABAQUS software to explore numerical models under diverse conditions, encompassing neat adhesive, 5% GBs, 10% GBs, and 15% GBs, while comparing these models with experimental data. The research content is comprehensive and logically organized. Furthermore, the references are aptly cited.
- Nevertheless, the article appears to present a superficial application of established XFEM technology, lacking a focus on crucial scientific issues within the adhesive field, and thereby failing to demonstrate innovation. Moreover, the poor clarity of Figures 1, 3, 14, 15, and 16 seemingly reflects a lack of meticulousness in the author's scientific approach. As a result, it is recommended to reject this article.
- The article was deeply revised taking into account the comments of the reviewers in terms of the information provided, the referred images were improved, and their explanation also better formulated. We believe also that we have made the innovation present in this work clearer.
- This study is a continuation of previous experimental investigations on experimentally changing the failure mechanisms of bonded SLJ from mixed failure to fully cohesive failure by adding hollow glass beads (GBs) to the adhesive mixture .
- From all this experimental work, done to evaluate the influence of 0, 5, 10 and 15\% of particles on two different adhesives, the purpose of the project was met, i.e., what was considered as full cohesive failure . However, only after an extensive amount of tests were performed. As such, in this work, a numerical model intent on reproducing the change in failure mode seen in experimental data was devised, in order to reduce the number of tests need to achieve the same conclusions for similar applications.
- This model was devised recurring to XFEM, but simulating not only the presence of the adhesive but also introducing the glass beads into the model, reproducing their interactions during the joint's failure. Contrarily to the common method that uses the cohesive properties of the doped adhesive to devise a CZM formulation requiring experimental tests for each configuration. Additionally this approach cannot simulate the filler-adhesive failure mechanics as CZM always assumes cohesive failure.
Reviewer 2 Report
Comments and Suggestions for Authors
In this paper, a numerical technique is proposed that can accurately predict the behavior of bonded joints, considering not only their failure mode, but also the strength of the joints when inorganic fillers are added to the adhesive. Moreover, different amounts of insulating glass particles were added to the epoxy adhesive to simulate the effect on the single lap test specimen. The numerical results are compared with the test results, and the joint strength and failure mode are analyzed. It has certain reference value.
However, there are several suggestions that need to be modified:
1. There are no quantifiable results in the abstract, which are basically qualitative conclusions, which cannot convince readers, please revise; 2. In line 204-206 of the paper, it is mentioned that two kinds of hollow glass beads with different proportions of 5% and 10% are used to conduct the experiment, while the experimental data of 15% hollow glass beads are repeatedly provided in the content of the paper and part of the figure. Is there any contradiction? It is suggested to adopt the consistent expression and experiment. Of course, adding 15% or more to the scale makes the results more useful;
3. The simulation results of Figure 16 in the paper do not add two different proportions of hollow glass beads of 5% and 10% for simulation, so it is recommended to add them, which will be more valuable;
4. The serial number of the equation in page 5 and page 6 of the paper is wrong, please correct it carefully;
5. Some figures in the paper, such as Figure 11,14,15 and 16, are not clear enough, so it is suggested to modify them.
Author Response
The authors would like to thank the reviewer for all the comments that helped this work become clearer and more well founded.
- There are no quantifiable results in the abstract, which are basically qualitative conclusions, which cannot convince readers, please revise;
- This was added to the manuscript both in the abstract and detailed better in the results section.
- In line 204-206 of the paper, it is mentioned that two kinds of hollow glass beads with different proportions of 5% and 10% are used to conduct the experiment, while the experimental data of 15% hollow glass beads are repeatedly provided in the content of the paper and part of the figure. Is there any contradiction? It is suggested to adopt the consistent expression and experiment. Of course, adding 15% or more to the scale makes the results more useful;
- Experimentally, only one type of spheres was used, in 4 quantities as seen in Figure 9 (0%, 5%, 10%, and 15%). The numerical study only considered 0%, 5% and 10% since in a previous paper the 10% proved to be the best ration between cohesive failure percentage and strength.
- The 15% results were removed to prevent confusion in relation to the work presented in this paper, this was better clarified in the manuscript.
- The simulation results of Figure 16 in the paper do not add two different proportions of hollow glass beads of 5% and 10% for simulation, so it is recommended to add them, which will be more valuable;
- Figure 16 was simply a representative image of the phenomenon seen in both 5% and 10% spheres, the stress distribution in the adhesive layer shows stress concentrations near the glass beads, making crack propagation go from the adhesive stress concentration (Figure 15) to the glass beads, as seen experimentally.
- Has seen experimentally the higher the sphere content the higher percentage of stress concentrations will be present in the adhesive, making the joint fail at lower loads. This makes the relation between configurations clear.
- A better clarification to this point was added.
- The serial number of the equation in page 5 and page 6 of the paper is wrong, please correct it carefully;
- This comment was not clear since no Equation was present in page 5 or 6. But the equations were revised and these errors corrected.
- Some figures in the paper, such as Figure 11,14,15 and 16, are not clear enough, so it is suggested to modify them.
- The figures were improved in the revised manuscript.
Reviewer 3 Report
Comments and Suggestions for Authors
This paper presents a numerical technique, which enables the precise prediction of the bonded joint’s behavior, regarding not only its failure mode but also the joint’s strength when inorganic fillers are added to the adhesive.
1. The introduction needs to be revised. The theoretical aspects of the work shall be presented in a separate section from than introduction.
2. The novelty of the work must be highlighted in the last paragraph of the introduction section.
3. As the main focus of the work is on cohesive laws and simulation of adhesive joints, I recommend you add more relevant recent publications to the introduction such as the below ones
Effects of nanoparticles on nanocomposites mode I and II fracture: A critical review
A Finite Element Model for Predicting the Static Strength of a Composite Hybrid Joint with Reinforcement Pins
Characterization of cohesive model and bridging laws in mode I and II fracture in nano composite laminates
4. What have you developed? A numerical approach or a theoretical model. What was the gap in the previous works?
5. All consumed materials in the research shall be presented in detail (Country, City, product code, referred to their datasheet, etc.).
6. In the experimental work, you are referring to previous works done by several researchers. Their results for the same materials could be different from what you have provided. Are these materials from the same supplier? Do they have the same grade? The same tested conditions. Explain.
7. Page 8, line 253: “partner company”. Who is the partner company?
8. The “2. Experimental Details” section needs to be rewritten. What you have done is not clear.
9. Add some images/graphics of the manufacturing process to the experimental section.
10. How did you control the thickness of the adhesive in the overlap area?
11. What standard test did you follow to carry out the tests?
12. How did you control and monitor the dispersion of GBs into the adhesive?
13. What about the size and density of the GBs?
14. How did you derive the traction-separation law for your adhesive?
15. How do the GBs affect the traction-separation law?
16. Did you do an ENF test to derive the traction-separation law for your adhesive?
17. Without knowing the values from ENF, how could you run the numerical model for predicting the shear strength of the samples? It is completely dependent on the experimental work.
Author Response
The authors would like to thank the reviewer for all the comments that helped this work become clearer and more well founded.
- The introduction needs to be revised. The theoretical aspects of the work shall be presented in a separate section from than introduction.
- These aspects about the XFEM formulation were moved to another section.
- The novelty of the work must be highlighted in the last paragraph of the introduction section.
- The novelty of this work, referred in comment 4, was better explained in the end of the introduction.
- As the main focus of the work is on cohesive laws and simulation of adhesive joints, I recommend you add more relevant recent publications to the introduction such as the below ones
- Effects of nanoparticles on nanocomposites mode I and II fracture: A critical review and Characterization of cohesive model and bridging laws in mode I and II fracture in nano composite laminates and Performance of various fillers in adhesives applications: a review and Influence of fillers on epoxy resins properties: a review
- We believe all of the references used were relevant to explain the whole picture of this project, but in light of your recommendation we added the work presented in the references above to the final manuscript.
- What have you developed? A numerical approach or a theoretical model. What was the gap in the previous works?
- A numerical approach to predict failure of joints doped with particles. It required extensive experimental testing to be able to predict the behaviour of our joints. This methods simply requires the characterization of the adhesive in the neat state, then the effect of the spheres is predicted by the simulation, resulting in a simple approach to test new joint configurations.
- This fact was better explained in the manuscript.
- All consumed materials in the research shall be presented in detail (Country, City, product code, referred to their datasheet, etc.).
- All materials were provided by the partner company, the glass beads were detailed in all these aspects but commercial names of the adhesive and substrates used cannot be divulged due to a Confidentiality agreement, other than that their nature and properties were provided nonetheless.
- In the experimental work, you are referring to previous works done by several researchers. Their results for the same materials could be different from what you have provided. Are these materials from the same supplier? Do they have the same grade? The same tested conditions. Explain.
- The work done previously was devised by our research group and in collaboration with the same partner company, and as such all of the materials were the same.
- Page 8, line 253: “partner company”. Who is the partner company?
- ArcelorMittal Global R\&D is the partner company, detailed in the co-authors and acknowledgments. It was added a better clarification of this fact.
- The “2. Experimental Details” section needs to be rewritten. What you have done is not clear.
- In this section only a simple explanation of the methods is described to contextualize the project, since the experimental work and data used was obtained in a previous paper. Therefore, the procedures and methods used for manufacturing are already extensively described in previous published papers.
- As such, to prevent any confusion to the fact that this work solely compared previous experimental results to the numerical procedure defined, the experimental details section was removed from the manuscript. Nonetheless, any relevant information for the understanding of the numerical study was added to the numerical details chapter.
- Add some images/graphics of the manufacturing process to the experimental section.
- We would gladly added them, but this became irrelevant to this manuscript since Section 2 (Experimental Details) was removed.
- How did you control the thickness of the adhesive in the overlap area?
- The specimens were manufactured in a mold with the help of calibrated spacers that ensured adhesive thickness. This became irrelevant to this manuscript since Section 2 (Experimental Details) was removed.
- What standard test did you follow to carry out the tests?
- These experimental SLJ specimens were manufactured and tested in accordance with standards ISO 4587:2003 and ASTM D1002-10(2019).
- How did you control and monitor the dispersion of GBs into the adhesive?
- The experimental procedure consisted of mixing the adhesive and the glass beads in a speed mixer at up to 3000rpm. This procedure has been tested previously in terms of particle distribution and agglomerate formation. These studies proved that this type of particle presents a homogenous distribution without agglomerates. Additionally, several SEM images presented in the previously published paper on this subject also show a uniform particle distribution. More details in the previously published paper on the experimental work.
- The numerical procedure defined a Matlab code that distributed the defined % of beads along the adhesive layer in such a way that their distribution was semi-random but approximately homogenous as a whole, similarly to the experimental observations. This code generated the geometry that was inputted into Abaqus.
- What about the size and density of the GBs?
- This information was missing and was added to the manuscript.
- How did you derive the traction-separation law for your adhesive?
- Using the Abaqus CAE functionalities of Energy-based CZM (Cohesive Zone Modeling). Were the traction-separation law (Figure 6) was obtained by the indirect method where the law is defined by 3 mechanically characterizable properties, for each mode. These properties were obtained through 4 different tests presented in the XFEM formulation section.
- A better version of Figure 6 was used to better relate the concepts.
- How do the GBs affect the traction-separation law?
- Since in this method the glass beads are simulated independently from the adhesive, each with its properties/law, there should not be any significant influence of the beads in the adhesive’s traction separation law. Then the XFEM method simulated the behaviour of the whole mixture.
- This fact was better described in the manuscript.
- Did you do an ENF test to derive the traction-separation law for your adhesive?
- Yes, in a previous study, as detailed in Table 1 of the original manuscript.
- This fact was better described in the revised manuscript.
- Without knowing the values from ENF, how could you run the numerical model for predicting the shear strength of the samples? It is completely dependent on the experimental work.
- The ENF tests were performed as previously described, these tests resulted in the GIIC property presented in Table 1, already mentioned before.
Round 2
Reviewer 1 Report
Comments and Suggestions for Authors
The scientific content of this article has been significantly improved and I believe the current version can be accepted after optimizing the format. Specifically, the format of Table 1 needs to be the same as Table 2 and 3 which are three-line tables.
Author Response
Table 1 was improved as suggested.
Reviewer 3 Report
Comments and Suggestions for Authors
The paper is accepted in its current form.
Author Response
No minor review comments were made, since the reviewer accepted the prior revision.